# A Comparison of Cats (*Felis silvestris catus*) Housed in Groups and Single Cages at a Shelter: A Retrospective Matched Cohort Study

**DOI:** 10.3390/ani8020029

**Published:** 2018-02-14

**Authors:** Malini Suchak, Jacalyn Lamica

**Affiliations:** Department of Animal Behavior, Ecology and Conservation, Canisius College, Buffalo, NY 14208, USA

**Keywords:** cat, *Felis silvestris catus*, animal shelter, group housing, social housing

## Abstract

**Simple Summary:**

When cats are relinquished to shelters, they frequently experience a great deal of stress. Shelters often try to control certain aspects of their environment, such as housing, to help them relax. Some cats are placed in small group rooms upon entry, whereas others are placed in single cages. There are tradeoffs to both systems. We wanted to compare the experience and outcomes of cats placed in single housing and group housing in shelters. We found that their experiences while at the shelter were similar, however single-housed cats were moved to the isolation unit more frequently than group-housed cats, which reduced their visibility to the public by pulling them off the adoption floor. Single-housed cats were also sent to the offsite location more frequently, which means they spent more time in transport than group-housed cats. Both groups were adopted at approximately the same rate and after similar lengths of time. However, the rate of return was high, especially in group-housed cats, and live release rate after return was lower than after a cat’s initial stay. More research is needed to know why this is happening and how to reduce returns after adoption.

**Abstract:**

The merits of various housing options for domestic cats in shelters have been debated. However, comparisons are difficult to interpret because cats are typically not able to be randomly assigned to different housing conditions. In the current study, we attempted to address some of these issues by creating a retrospective matched cohort of cats in two housing types. Cats in group housing (GH) were matched with cats in single housing (SH) that were the same age, sex, breed, coat color, and size. Altogether we were able to find a match for 110 GH cats. We compared these two groups on several measures related to their experience at the shelter such as moves and the development of behavioral problems. We also compared these groups on outcomes including length of stay, live release, and returns after adoption. We found that while the frequency of moves was similar in both groups, SH cats were more likely to be moved to offsite facilities than GH cats. SH cats also spent a smaller proportion of time on the adoption floor. Length of stay and, live release and returns after adoption did not significantly differ across groups, however GH cats were two times as likely to be returned after adoption. Future research should look at the behavioral impacts of shelter decision-making regarding moving and management of cats in different housing systems.

## 1. Introduction

With approximately 3.2 million cats entering U.S. animal shelters every year [1], there is an essential need for shelters to develop housing systems that are efficient and cost effective, while at the same time maximize the welfare of the cats during their stay. Several housing features that help improve welfare are widely agreed upon. First, properly sized enclosures, that allow the cat space between feeding, resting, and elimination areas, are important [2,3]. Second, predictable and stable environments are important for reducing stress and the expression of sickness behaviors [4,5]. Finally, enrichment is important on a variety of levels. Structural enrichment, such as shelving providing elevated vantage points or proper hiding spots, is important for the cats to feel secure [6]. Social enrichment, in the form of interactions with conspecifics or people is important for most cats, however, there is individual variability due to different experiences during early socialization periods [7]. Altogether, two major housing systems have emerged with the potential to provide most, if not all of these key features, group housing (GH) and single housing (SH).

Group housing typically consists of a few to many cats housed in a large room that is furnished with beds, shelves, and other enrichment. In previous studies of group housing the number and density of cats in the room (cats per m^2^ of space) is highly variable, indicating highly inconsistent practices across shelters in this regard [8,9,10]. Nevertheless, recommendations usually suggest limiting the room to no more than 10 cats, or 1.67 m^2^ of floor space per cat, whichever is fewer [2,3]. The environment of group living often provides cats with more enrichment due to having more space in which to move around and more choice and control about how they use their environment. This may be why more solitary play and greater overall activity is observed in larger rooms [11]. It is not only the room size that results in more activity and playing, the presence of other conspecifics could play a key role in the increased activity of individuals in GH [12]. In a study by Uetake et al. [12], researchers found that there was more activity and play within GH, even if the SH cage was large enough for them to play in [12]. The authors suggest that space is not enough, but rather contact with conspecifics and with humans can increase normal behavior such as playing and reduce inactivity.

Given that inactivity in cats may be a sign of stress or the development of sickness behaviors, increased activity may be an indicator of positive welfare in shelter cats [4,5]. Play is also often used as an indicator of positive welfare, since play is thought to only occur if the animal’s needs are met and the animal feels secure from threats [13,14]. Furthermore, greater activity, solitary play (especially play with toys), and the presence of enrichment may positively influence adoption outcomes in any housing setting [12,15,16]. Activity has been shown to draw the attention of potential adopters and result in them viewing active cats for a longer period of time than inactive cats [15]. Although these things are certainly not unique to GH, they appear to be better facilitated in larger rooms with conspecifics present [11,12].

However, the nature of GH is often at odds with the natural history of the cat. Cats do not have a long evolutionary history of social living, and free ranging cats that aggregate tend to live in relatively stable groups of related females, with the males dispersing from their natal groups [7,17]. In shelters, groups are inherently unstable, with compositions changing as individuals get adopted, and often consist of unrelated males and females together. Further, whether a cat benefits from GH is also dependent on the sociability of each individual, the environment that they were reared in, and whether or not they are placed with relatives, familiar, but unrelated cats, or strangers [6,18]. For example, in a preliminary study of 25 cats moved to a boarding cattery for five days, Bradshaw and Hall [18] found littermates were more likely to spend time in proximity and showed more affiliative behaviors than non-littermates. However, in our own work, examining 259 cats over their entire length of stay at the shelter, we did not find any difference in proximity between familiar and unfamiliar pairs of cats [19].

Altogether, whether or not a cat is suited to living socially appears to be highly individually dependent [7]. If an individual is not comfortable with other conspecifics he or she might be less inclined to play and more inclined to hide or avoid others [18]. In such a case, SH is clearly the better option. Although it may seem as if it is more difficult for SH to provide the enrichment and stimulation that a larger, group setting might provide, there are a number of methods to increase activity and provide enrichment even in smaller cages, especially by creating double cages with “portholes”, which allow for the separation of an area for eating, sleeping, and playing from an elimination area [2]. While GH may provide cats with better hiding opportunities, which allows them to better cope with stress, a hide box easily provides them with this opportunity in SH [20]. Indeed, with proper enrichment, adoption outcomes such as length of stay are similar in SH and GH [16].

There are other advantages to SH, such as social stress reduction and disease management. SH systems are typically set up in ways that make them easy to sterilize, ideally prevent the spread of disease between cats, and facilitate the monitoring of each individual cat. However, the advantages of this in terms of reducing illness is often assumed rather than substantiated and careful observation in GH cats can achieve the same result [21]. The advantage in terms of social stress is also not straightforward, due to variability in the sociability of individual cats to conspecifics, where some cats might prefer the company of other cats [9,22]. Obviously, if an individual is socialized to humans but not to other conspecifics, then single living would be less stressful for that cat [7,9,22].

It is clear that both settings have advantages and disadvantages in terms of maximizing the welfare of shelter cats. Several studies have sought to compare cats housed in different settings, typically using behavioral measures of stress such as the Cat Stress Score (CSS) or a physiological measure such as cortisol to creatinine ratio [8,9,12,22,23,24]. Results of these studies have been mixed. Some studies have found no difference in behavioral signs of stress or cortisol in group-housed versus single-housed cats [12,23,24]. Similarly, there was no difference in glucocorticoid metabolites in cats from single cat homes vs. multi-cat homes [25]. However, other studies have found signs of higher stress in group housing, although this seems specifically related to higher density, rather than the housing system per se [8,9,26] and one study using infrared thermal imaging of core eye temperature to measure stress found higher stress levels in singly housed cats [27].

Finka et al. [10] critically analyzed a number of these studies and found that the mixed results could be due to many reasons including different methods of assessing stress, confounding variables including moves and time spent in transport, and a lack of information provided in studies, especially as pertaining to the husbandry procedures and handling of the cats. One reason why there may be no clear answer is because cats in any housing system experience stress, but from different sources and have different options for coping. Thus, instead of one system maximizing welfare, there may be tradeoffs experienced in both systems. Furthermore, given that density is frequently related to stress, key information is often missing including the overall number of cats, and the size and spatial features of the rooms. Altogether, this makes it quite difficult to evaluate comparisons between groups.

An additional complication of comparing cats in single and group housing is that the groups may not be equivalent to each other, and relatively few studies have sought to control for the characteristics of the cats themselves and the treatment of each group at the shelter. For example, Brown and Morgan [28] found that age and physical features of the cat (like coat color) influence length of stay (LoS). In turn, LoS may influence stress and the development of illness [29]. To our knowledge, only one study has attempted to control for the characteristics of the cat [16] and it is unclear how systematically that was done. Furthermore, it is unclear, as mentioned above, if cats in both settings were moved with equal frequency and spent similar amounts of time in transport. Differing experiences between singly housed and group-housed cats might influence their welfare and stress while at the shelter.

In the current study we sought to examine whether the cats in single and group housing have similar experiences in the shelter and similar outcomes. Behavioral measures of stress were not focused on in this study for the following reasons. First, although widely used, there are inconsistencies between CSS and hormonal measures, which may be a result of CSS underestimating stress in inactive cats [30]. Second, hormonal measures are useful, but impractical on a larger level as they are cumbersome and expensive, and, in group housing, often difficult to obtain without disrupting the cats. Third, to our knowledge, there have been no studies that have examined other aspects of the cat’s experience that might influence stress and their outcomes. As mentioned above, these experiences might highlight the tradeoffs cats face in different housing systems.

Unlike most previous studies, we created a retrospective matched cohort of cats in each housing system by matching individuals based on the features of the cat. We explored their experiences in two key areas: shelter procedures (i.e., moving the cats between cages and locations) and outcomes (such as length of stay, live release rates, and returns after adoption). Given the conflicting information in the existing literature comparing housing systems, and the lack of information on factors other than stress, we did not form directional hypotheses for any measure. Rather, we consider this an exploratory study to examine whether there are differences in housing systems at one shelter when the physical features of the cats are controlled for.

## 2. Materials and Methods

### 2.1. Subjects and Setting

220 adult cats who were housed at the SPCA Serving Erie County in Western New York state during 2014–2015 were the subjects for this study, 110 of whom were placed in single housing (SH) and 110 in group housing (GH). The SPCA Serving Erie County is a managed admissions shelter, which accepts cats on an appointment-based system as room allows, and does not euthanize for behavioral or space reasons. Immediately after intake, cats undergo a medical examination. Cats who are deemed healthy were typically immediately placed on the adoption floor, or into a holding area until a space on the adoption floor opened up. Typically, holding only lasted 1–2 days prior to the cat being moved to the adoption floor. Groups were formed using an all-in, all-out procedure. Cats were placed into either single housing or group housing at the discretion of the staff and at no time did the researchers play a role in this decision.

SH cats were housed in standard metal cages ranging in size from 0.38 m^2^ of floor space (dimensions in cm: 54.61 w × 54.61 h × 71.12 d)–0.85 m^2^ of floor space (in cm: 119.38 w × 71.12 h × 71.12 d). Most cages had two compartments with a porthole, although some did not. Assignment to particular cages depended on the size and activity level of the cat and what was currently available. GH cats were housed in one of four colony rooms in groups ranging from 2–8 individuals (median: 3). Colony rooms had 3.06–5.41 m^2^ of floor space, leading to a median density of 1.02–1.80 m^2^ of floor space per cat (Figure 1).

Both SH and GH received dry food and water ad libitum, as well as two daily feedings of wet food, as per the shelter’s husbandry procedures. Spot cleaning, including litter changes were performed for all cats in every housing setting daily. Additionally, each day a veterinary technician visually examined every single cat to check for health concerns. Both SH and GH cats also had beds, towels or other soft resting places and toys available. SH cats did not have a hide box or perch provided. Additionally, GH cats had shelves, windows that either looked outside or into the lobby, a kuranda tower, and often other enrichment such as milk crates or boxes. Both SH and GH cats received anywhere from 15–30 min of human interaction (playing, petting, socialization) on days the shelter was open, as per the practices of the shelter’s volunteer cat enrichment team. All husbandry and enrichment was implemented as per the shelters procedures and the researchers played no role in these decisions. Shelter procedures did not substantially change during the study period.

Cats in this study were categorized as SH or GH based on their housing while on the adoption floor. In addition to their time in SH or GH housing, there were several other possible housing locations for cats during their time at the shelter that were not on the adoption floor. Sick cats were sometimes pulled to the isolation unit, which consisted of cages similar to those found in SH. Both SH and GH cats could be moved to the isolation unit, but typically GH cats were returned to group housing on the adoption floor following their stay. Movement into isolation was at the sole discretion of the veterinary staff and was decided based on the threat to the cat’s own health, the necessity of intensive treatment, as well as the impact a removal might have on the rest of the group. There were also several offsite locations in mall or pet stores. These contained either standard SH caging, “cat playpen” caging, approximately 1.16 m^2^ of floor space (in cm: 91 h × 59.69 d × 128.17 w) wire condo units which typically contained vertical levels, or group housing. Cats could also be sent to foster homes if they were stressed by the shelter environment or needed extra socialization prior to adoption. Cats were also occasionally located in holding cages out of the public view if they were waiting for a room or cage placement, or if they were a bite/scratch case on a ten-day rabies hold.

### 2.2. Method for Matching

Since fewer cats moved through group housing than single housing, we used a list of 259 adult subjects that were housed in groups in the colony rooms during 2014–2015 period as our starting point. These cats were compiled through a previous, unrelated behavioral study examining proximity between cats in group housing [19]. For each cat we recorded their age upon intake, sex (male/female), primary breed, primary coat color, and size from the PetPoint database. We also eliminated any cats for whom this was not their first stay at the shelter, confining our subject pool to only the initial intake for each cat. To obtain potential matches, we used the animal inventory reports to pull all of the cats who moved through SH during the 2014–2015 period. We then eliminated any cat who was housed with another cat at any time from this list and any cat that moved from SH into GH during their time at the shelter. That left us with a potential list of 637 cats. We used the previously identified features (age, sex, primary breed, primary coat color, size) to identify matches; a SH matched cat had to share all of those features with the corresponding GH cat. Only one SH match per GH cat was allowed, if multiple matched the cat with the closest weight and physical appearance was used.

We were unable to precisely match for intake date or season as this would have led to a prohibitive reduction in potential subjects, however, we ensured relatively equal numbers of cats in SH and GH across the two years (56.36% of SH cats and 55.45% of GH cats were adopted in 2014), and across the slower, winter season and the busier, summer season when there were also a lot of kittens present at the shelter (52.72% of the cats from each group were adopted in the summer, with the remaining 47.28% adopted in the winter).We were unable to match on spay/neuter status since most cats are admitted intact, but then are spayed or neutered at various points throughout their stay, such as immediately prior to adoption or move to an offsite location. Given that our measures look at a cat’s entire length of stay at the shelter it would be impossible to classify cats as fixed or intact since their status changed throughout their stay and, at most, a spay/neuter surgery lengthened a cat’s stay by one day. Altogether using this process, we identified matches for 110 of the group-housed cats, creating a total pool of 220 cats (110 from each housing system) to examine. We used G*Power to perform a sample size analysis [31] using a medium effect size (d = 0.5 for the Wilcoxon rank-sum tests and w = 0.3 for the chi square), an alpha of 0.05, an anticipated power of 0.95, to determine that 110 individuals in each group, and 220 total would provide adequate power.

### 2.3. Shelter Experience Measures

Aside from the obvious difference in terms of cage size and structure, we were interested in whether SH and GH cats move through the shelter system in a similar way. We used several measures, obtained from information logged in the PetPoint database for each cat:(a)Number of moves: The number of times a cat was moved from one cage or room to a different cage or room and remained in the new cage or room for more than 24 h. Therefore, a short-term move (for example, back to the clinic for a quick physical) was not counted. Given that moves are considered to be a stressful event for cats [21,23,32], differences in the number of moves between SH and GH cats could indicate the potential for higher stress in one group.(b)Average time spent in each location: Cats who were present longer at the shelter may experience more moves due to the overall amount of time. Therefore, we calculated the average time spent in each location as their length of stay (the number of days between their intake date and their outcome date, see below) divided by their number of moves (measure (a) above). This gives a measure of the average number of days per location, or the frequency with which the cat was moved.(c)Length of time on adoption floor: Since some of the moves (such as a move to the isolation unit, foster, or bite/scratch case) might result in a cat being moved off of the adoption floor, we looked at length of time on the adoption floor separately from these other measures. Time on the adoption floor has two implications for the cat. First, in order to be adopted, a cat typically needs to be visible to the public [15]. Second, the adoption floor tends to be much busier, with a great deal more human interaction than other areas.(d)Offsite moves: If a cat was marked as moving to an offsite location at any point during their length of stay, we categorized them as having been moved offsite. This was a 1/0 measure; cats who were moved offsite were scored as 1, cats who were not were scored as 0. Offsite moves increases time spent in transport, which is thought to cause stress [32].(e)Foster moves: If a cat was moved into a foster program, we marked them as moving to foster. This was a 1/0 measure; cats who were moved to foster were scored as 1, cats who were not were scored as 0. Foster moves also increases time spent in transport, since the cats are moved to the foster home and then back to the shelter for adoption. We excluded “foster to adopt” cats from this category if they were ultimately adopted into those same homes (and therefore did not return to the shelter for adoption). The reason for that is that in every case for our subjects, foster to adopt was used when the adopter was unable to officially adopt the cat until the cat finished a round of antibiotics and was cleared by the veterinarian for insurance reasons, which typically occurred 2–3 days after the cat became “foster to adopt”. Therefore, “foster to adopt’ was an artifact of an insurance policy and was qualitatively quite different from the cats needing true foster care for a period of time before being returned to the shelter for adoption.(f)Bite/scratch cases: Cats occasionally developed or exhibited behavior problems while in the shelter. The most severe of these were bite/scratch cases, as each time the cat was reported as biting or scratching someone, he or she would have to be held for a ten-day rabies watch without contact from visitors or volunteers. This has serious implications for the cat’s stay at the shelter including length of stay, enrichment, and housing setting. This was a 1/0 measure; cats who were bite/scratch cases were scored as 1, cats who were not were scored as 0.

### 2.4. Outcome Measures

(a)Length of Stay (LoS): Length of stay was calculated as the entire time spent at the shelter from intake to outcome. Dates were determined by official dates logged in the PetPoint system upon admission to the shelter and outcome.(b)Live release: A cat was considered live release if he or she was adopted, returned to the original owner, or transferred to a private rescue or barn cat program. Cats that were euthanized or died while at the shelter due to other causes were not considered to be live release. Live release was a 1/0 measure; cats who were released alive were scored as 1, cats who were euthanized, or otherwise died while at the shelter were scored as 0.(c)Returns following adoption: For cats that left the shelter through a positive outcome (adoption or return to owner/guardian), we looked to see if the cat was returned within two years of adoption. Return was a 1/0 measure; cats who were returned in the two-year period were scored as 1, cats who were not returned were scored as 0. 

### 2.5. Analysis

Our independent variable throughout was single vs. group housing. Each measure listed above was treated as a separate dependent variable. However, some of these measures were related to each other (see Appendix A). Since the continuous measures (number of moves, average time spent in each location, length of stay) were not normally distributed, non-parametric Wilcoxon rank-sum tests were used. For the same reason, medians are also reported instead of means as the measure of central tendency. We also had several categorical dependent variables (offsite moves, foster moves, bite/scratch cases, live release, returns). Since each of these variables was a 1/0 measure, we used a 2 × 2 chi square test for each measure and calculated the odds ratio of each outcome for SH cats relative to GH cats. Alpha was set to *p* = 0.05. All analyses were done using SPSS for Macintosh, version 24.0 (IBM Corp, Armonk, NY, USA).

## 3. Results

### 3.1. The Cat’s Experience in the Shelter

Results of the experience measures are summarized in Table 1. Housing type did not impact the number of moves and average days spent in each location. SH cats (N = 110) were moved a median of 2 times (interquartile (IQ range): 1–4 times), and spent a median of 10.41 days in each location (IQ range: 7.00–15.63 days). GH cats (N = 110) were also moved a median of 2 times while at the shelter (IQ range: 1–3 times), and spent a median of 12.71 days in each location (IQ range: 7.00–20.00). There were no significant differences between groups on either measure (number of moves: W = 11,567.50, Z = −1.27, *p* = 0.21; average days per location: W = 12823.50, Z = 1.42, *p* = 0.16) However, SH and GH cats were often being moved to different locations. Specifically, SH cats (N = 110) were more than 2.5 times more likely to be moved offsite than GH cats (N = 110; SH: 67.27%, GH: 44.55%, χ^2^ = 11.52, *df* = 1, *p* < 0.001, OR: 2.56, CI: 1.48–4.43, Figure 2).

There were also several circumstances in which the cat might be moved off the adoption floor and out of sight of the public. We found that proportion of days the cats were visible on the adoption floor significantly differed between the two groups. SH cats (N = 110) tended to be available for a median proportion of 0.67 of the time (IQ range: 0.56–0.89), whereas GH cats (N = 110) tended to be available for a median proportion of 0.97 (IQ range: 0.78–1.00; W = 14844.00, Z = 5.78, *p* < 0.001, Figure 3). It is important to note that GH cats had very strong skew on this measure. The vast majority of them were visible nearly their entire length of stay, however, there were a number of extreme outliers available for a very small proportion of their stay. They key driver of this difference appears to be moves to the isolation unit. SH cats were 3.2 times more likely to be moved to isolation as GH cats (with 30.00% of SH cats moving to the isolation unit, but only 11.82% of GH cats; χ^2^ = 11.00, *df* = 1, *p* = 0.001, OR: 3.20, CI: 1.58–6.49). Given that the median time spent in the isolation unit was 9 days (IQ range: 8–13, N = 46), moves to isolation likely had a serious impact on the time spent on the adoption floor. In contrast, only 4 cats were moved into foster care, which typically would result in a long-term removal from the adoption floor. Although all four were GH cats, the rate of occurrence (1.1% of cats) is so low it is impossible to tell if this is a coincidence or a larger pattern. Finally, very few cats (SH: N = 6, GH N = 9) developed a serious behavior problem such as biting or scratching, which would also require removal from the adoption floor. The proportion was relatively evenly distributed across groups and cats in SH and GH were equally likely to be classified as a bite/scratch case (*χ*^2^ = 0.64, *df* = 1, *p* = 0.42, OR: 0.65, CI: 0.2–1.89) (Table 2).

### 3.2. Outcomes

Results of the outcome measures are summarized in Table 2. The median length of stay was very similar for both groups, and indeed there was no significant difference in the length of stay for SH and GH cats (SH median = 24.50, GH median = 26.00, W = 12,416.00, Z = 0.55, *p* = 0.58, N = 110 in each group). Similarly, there was no significant difference in the live release rate for both groups (SH = 96.36%, GH = 92.73%, χ^2^ = 1.41, *df* = 1, *p* = 0.24, OR: 2.08, CI = 0.61–7.12). We compared the proportion of cats returned to the shelter within two years. Only cats that were adopted or returned to their owner could be returned to the shelter, resulting in a comparison of 104 SH cats to 93 GH cats. Cats that were transferred out or euthanized were excluded as they could not be returned to the shelter. In this group, SH cats were 0.54 times as likely to be returned as GH cats, but this difference was not significant (SH return rate: 11.54%, GH: 19.35%, χ^2^ = 2.32, *df* = 1, *p* = 0.13, OR: 0.54, CI = 0.25–1.20). It is important to note, however, that 25 out of the 30 (83.33%) cats that were returned were adopted into different homes, and, are still in those homes to the shelter’s knowledge.

## 4. Discussion

The two most common housing systems for shelter cats, single housing and group housing, represent tradeoffs in efficiency for the shelter and welfare for the cats. Numerous studies have attempted to quantify the impacts of these housing systems on the welfare of cats, often by using a behavioral or physiological measure of stress [8,9,12,22,23,24]. These studies have yielded mixed results, and some researchers have suggested that more information about confounding variables associated with housing type, such as differences shelter management procedures is important to provide context for behavioral comparisons [10]. In this study, we sought to compare SH and GH cats on measures relating to their management in the shelter and outcomes. Information about systematic differences in these areas can provide context for other studies and prompt consideration of previously overlooked external variables that might influence stress.

To our knowledge, this was also one of few studies that systematically controlled for so many of factors, including age, sex, and coat color, that are known to impact length of stay and other aspects of the cat’s time at the shelter [28]. Given that practically and ethically it is difficult to randomly assign cats for the purposes of research, we propose that post hoc analyses using the vast array of data stored in shelter databases is a promising method for further research in this area. It could also be extended to behavioral studies, where large bodies of behavioral data are amassed and then post hoc cats can be matched based on age and physical characteristics.

Our measures of the cat’s experience in the shelter largely reflected decisions by the staff and interactions between the cat and people. Although in some cases there is no a priori reason to assume that cats might be treated differently, on a practical level, one might imagine that SH cats are easier to move and manage, since each individual move does not involve the formation or dissolution of a social group. Group stability is often emphasized as important [3,10], which may lead to an undue burden on SH cats for moving, since moving an SH cat from one cage or location to another does not impact the rest of their social group. We found that despite this practical difference, SH cats were not moved more frequently or to a wider number of cages than GH cats. Given that most of the GH cats in our sample were not admitted to the shelter already socially bonded to other cats [19], it may have been assumed that they could be moved just as easily as SH cats. As a next step, it would be important to know whether these moves of GH cats have implications for the other cats in their social group. Although best practices suggest that groups should be kept as stable as possible, cats moving out of social groups is an inevitable part of shelter life and proper space can help reduce stress and conflict [33]. This would require behavioral study, beyond the information that a database survey might provide.

When they were moved, SH cats were 2.56 times more likely to be moved to an offsite location, whereas GH cats more often tended to be moved to another location within the shelter. This means that SH cats spent more time in transport than GH cats. This may lead to increased stress in SH cats, since it appears to take cats at least 24 h to acclimate to a new setting [4,34], however, the current study cannot speak to the impacts of that. Further, it may be unavoidable as the offsite locations are in areas with a lot of foot traffic and can be useful for moving cats out of the shelter efficiently. Thus, they play an important role in the shelter ultimately being able to admit more cats. It is important to note however, that nearly half of all GH cats do get moved to an offsite location at some point as well, so for both groups it is not a rare occurrence.

Since SH cats might spend more time in transport, it seems that effort should be made to minimize moves while at the shelter. However, that already seems to be the case in our shelter as the median number of moves while at the shelter was just two moves for all cats. This number may be artificially low, since moves were limited to locations where the cats spent more than 24 h. Future behavioral studies should examine shorter-term versus longer-term moves, as both may be equivalently stressful for the cat.

Most concerning, however, are moves that may result in a cat being pulled off of the adoption floor, as cats who are not visible or available to the public typically are not adopted. We found that SH cats spent significantly less time on the adoption floor than GH cats, relative to their total length of stay. This effect largely seems to be driven by moves to the isolation unit, as SH cats were three times more likely to be moved to isolation. We believe this result demonstrates one of the tradeoffs shelters face when dealing with group housing, which is the best method for disease control. If a cat gets sick while in group housing, the shelter is faced with the decision of removing the cat to quarantine, possibly disrupting the group and causing stress to all of the cats involved, versus keeping the cat in a group where the disease may spread. Alternatively, if the cat is stressed out by the group dynamic, this could exacerbate the incidence of disease. Since one of the common diseases in shelters is Upper Respiratory Infection (URI) [29], which seems to be activated by stress [35,36], group housing presents an additional consideration when managing URI and other infectious diseases.

We analyzed two other reasons why cats might be pulled off the adoption floor. There was no significant difference in the number of bite/scratch cases across both groups and, in fact, the incidence of such serious behavior problems was rather low overall. The number of cats moved to foster homes was too small to analyze, however the only cats moved into foster care were GH cats. Three were for problems with socialization to humans and one was for a medical treatment who was returned to the shelter after treatment for regular adoption (and therefore was not foster to adopt). It seems odd that GH cats might have a tendency towards poorer socialization with humans, as little is known about the socialization status of any cat entering the shelter. It is possible that in the current setting, those problems are more recognizable in the GH setting because the cats have a greater opportunity to hide and escape from people. If single-housed cats were provided with a hide box, we might expect to see an increase in reported socialization problems in that setting as well.

The proportion of time spent on the adoption floor can also be interpreted relative to the cat’s overall length of stay (LoS). LoS was very similar across groups, with both SH and GH cats having a median stay of just over three weeks. Gourkow and Fraser [16] found that cats from single enriched housing had similar adoption rates to cats from group housing within 21 days of admission. However, although our single housing tended to have a bed, some toys, and often multiple compartments, we were missing a key feature of their single enriched housing, a hide box and perch. It is difficult to compare across studies, however, as LoS appears to vary widely between shelters and cats [28,37], and the full length of stay is not often reported in studies that look at adoption within a certain time frame (such as 21 days, [16]). Furthermore, our study highlights an often overlooked measure related to LoS, which is visibility on the adoption floor. Although the overall LoS was similar, SH cats spent significantly less time on the adoption floor, indicating that when they were visible to the public, they were getting adopted faster than GH cats. This could be due to the fact that more SH cats were moving to offsite housing, where they tend to be more visible to the general public.

Overall, the outcomes of the cats from both groups were extremely similar. The shelter’s live release rate was extremely high, which, given the managed intake during times when the shelter is full, and participation in a barn program for feral or unsocialized cats, is to be expected. However, approximately one in 10 SH cats were returned, and for GH cats that number was one in five. It is important to note that although the numbers for GH cats are rather concerning, the difference between SH and GH cats is not significant. Only 83% of the returned cats were adopted by new owners, so returns ultimately did decrease the live release rate for these cats. Unfortunately, the current setup of the database gives shelter staff an open text box to describe why the cat was returned, which leads to great variability in the reason entered, and prevents a quantitative analysis of the reason for return. However, looking at the qualitative descriptions, behavior was mentioned eight times for GH cats, but only one time for SH cats. Specifically, five of those eight times were for behavior towards another pet. Cats who are adopted from social housing may be viewed as more amenable to getting along with other pets, especially other cats. In one study, 69% of people said a cat being “friendly” with other cats was important to their choice [16] and the most common reason for return in another study was aggression between cats in the household [38]. Presence in group housing may be leading people to assume the cat is tolerant towards other cats. Adopters may also perceive a lower need to do a gradual introduction since it is assumed the cats are more sociable. This can lead to the failure to establish a friendly relationship, and in households where this happens, aggression can persist for years [39,40]. More systematic data collection upon return and perhaps interviews with adopters and owners returning pets would shed light on what is causing this problem.

Although we propose that the method of matching the cats in age and physical characteristics, and the finding that cats largely have similar experiences and outcomes regardless of housing situations are important contributions to the literature, there are, of course, some limitations to the current study. First, we only had access to data from one shelter, with very specific intake policies. Future research should examine data from other shelters and rescues, to see if these patterns persist. Second, some previous work has called into question the validity of information present in shelter databases [41]. Although there is always the risk that shelter personnel entered information incorrectly, there is likely a vast amount of valid data contained within shelter databases that is currently not being leveraged to improve outcomes and welfare in the shelter setting. Further, one of the key issues identified in previous work was cats not being physically present in the location specified on the database. As part of our broader research agenda examining hundreds of cats, we have found this only to be true for cats pulled for short-term procedures, such as spay/neuter surgeries (at which point they might still be listed as on the adoption floor, but not physically present in the room at the time) [42]. By limiting our analysis of moves to only those where the cat is in a location for 24 or more hours, we avoid this problem. Third, our study was conducted in situ, that is, in a fully operational shelter setting, without altering the procedures of the shelter or the housing systems for the purpose of the study. Although this provides a level of naturalness and authenticity to the results, we were unable to control for some features as closely as a more controlled laboratory study might. For example, our SH cage sizes ranged from 0.38–0.85 m^2^ of floor space because that is the pre-existing caging available at this shelter. Obviously it would be best if all cages were the same size and set up for standardization purposes. However, it is to the benefit of the field to have a combination of naturalistic and highly controlled studies in the literature. If studies are only done in idealized or tightly controlled laboratory settings, applicability to shelters would be limited.

There were also some limitations to our analysis, namely that, due to very high skew in many of our variables we were unable to do more complex regression modeling without individual cats unduly influencing the results. This also means that several of our dependent variables are related to each other, especially those that ultimately relate to the amount of time the animal is present in the shelter (see Appendix A). Therefore, it is important to interpret these measures in conjunction with each other. We also note that because the analysis of returns was limited to only cats with positive outcomes (that is, adoption or return to owner) this analysis may have been under-powered, leading to a potential type II error.

Finally, although shelter data relating to procedures and outcomes are very informative, ideally in future research this would be complemented by behavioral observations of cats in both housing systems. At this point, the impact of moves and returns on the cats is speculative, and systematic behavioral observations would be needed to better understand the impact on welfare. However, we believe the information from the database was useful in identifying a few key points where behavioral observations would be most useful, for example, relating to offsite moves or moving cats into or out of social groups and thus allowing researchers and shelter staff to focus on particular questions that may lead to the greatest benefit.

## 5. Conclusions

In conclusion, through the use of a retrospective matched cohort, we were able to compare SH and GH cats on several measures that have not previously been examined. Although we have several null findings, these are also informative. There are often assumed differences between the housing systems that do not pan out when systematically studied and when features of the cat such as age and coat color are controlled for. However, there are other differences that do need to be taken into account such as differences in transport rates and time spent on the adoption floor. Furthermore, our research identified one key area of concern, which is the high rate of returns. Shelters are often focused on initial live release rates, but the lower live release of returned cats plus the other welfare issues associated with returns make this an important problem deserving of future research. In theory, a retrospective matched cohort comparison could be extended to behavioral and welfare assessments as well. There is currently a great deal of conflicting literature on these topics, and more attention to controlling for the features of the subject pool may be essential to drawing conclusions that promote the better welfare of cats in the shelter setting.

## Figures and Tables

**Figure 1 animals-08-00029-f001:**
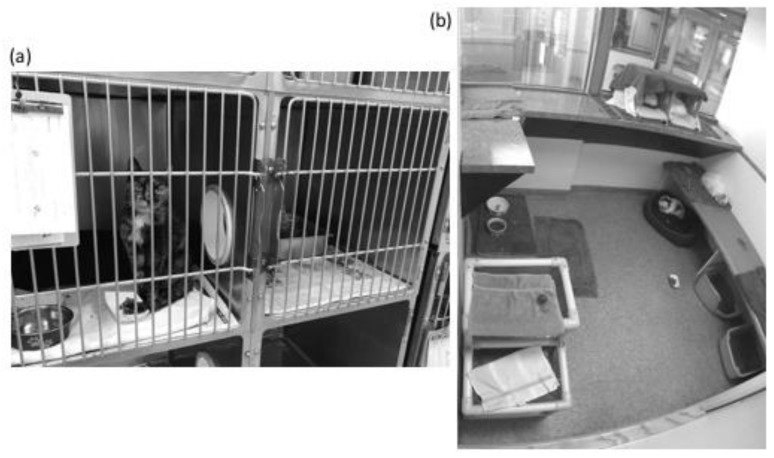
Example housing units. (**a**) A single cat cage with two compartments connected by a porthole; (**b**) A group housing room containing four cats in 5.41 m^2^.

**Figure 2 animals-08-00029-f002:**
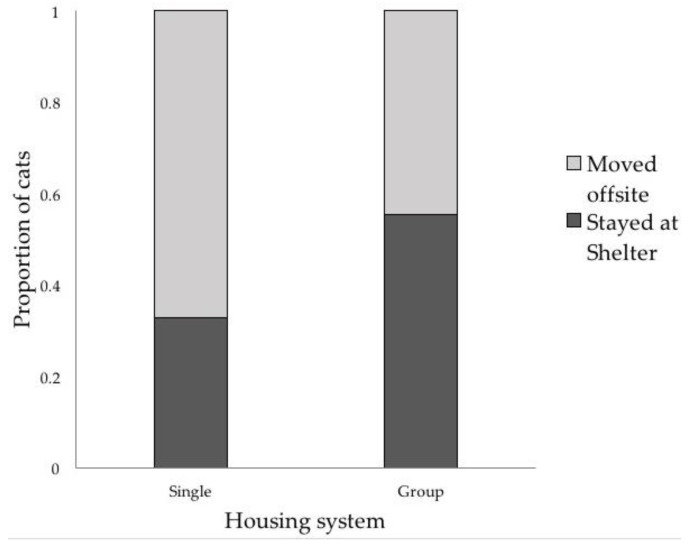
Single-housed cats were 2.5 times more likely to be moved offsite than group-housed cats (χ^2^ = 11.52, *df* = 1, *p* < 0.001, OR: 2.56, CI: 1.48–4.43).

**Figure 3 animals-08-00029-f003:**
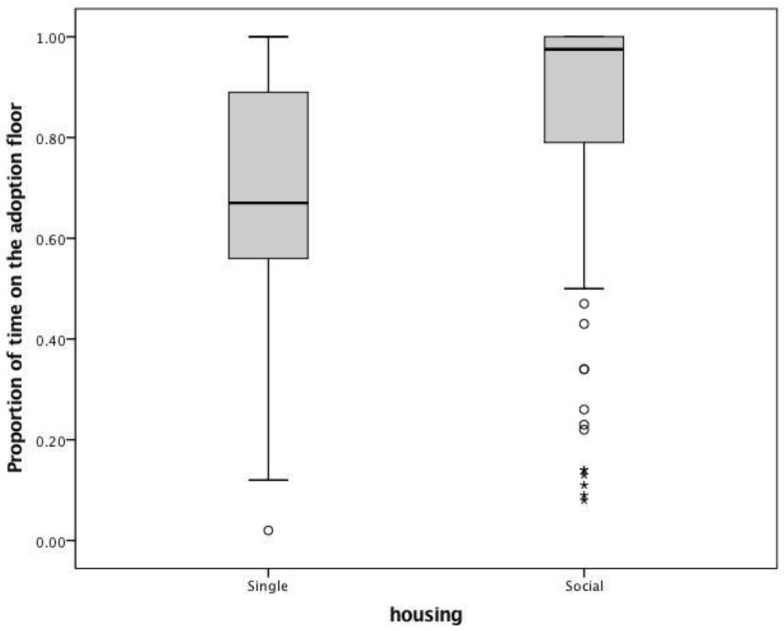
Single-housed cats spent significantly less time visible on the adoption floor than group-housed cats (W = 14,844.00, Z = 5.78, *p* < 0.001).

**Table 1 animals-08-00029-t001:** Summary of measures relating to the cat’s experience. Single-housed (SH) cats spent less time on the adoption floor, and were move likely to be moved offsite and to the isolation unit than group-housed (GH) cats.

**Continuous Variables**	**N**	**Median**	**Statistic**	***p*-Value**	**-**	**-**
Number of moves	SH: 110 GH: 110	2.00 2.00	W = 11,567.5	0.21		
Average days per location	SH: 110 GH: 110	10.41 12.71	W = 12,823.5	0.16		
Proportion of days on adoption floor	SH: 110 GH: 110	0.67 0.97	W 14,844.00	<0.001		
**Categorical Variables**	**N**	**Percent**	**Statistic**	***p*-Value**	**Odds Ratio**	**95% CI**
Offsite moves	SH: 110 GH: 110	67.27 44.55	χ^2^ = 11.52	0.001	2.56	1.48–4.43
Isolation moves	SH: 110 GH: 110	30.00 11.82	χ^2^ = 11.00	0.001	3.20	1.58–6.49
Bite/scratch cases	SH: 110 GH: 110	5.45 8.18	χ^2^ = 0.64	0.42	0.65	0.22–1.89

**Table 2 animals-08-00029-t002:** Summary of outcome analyses. There were no significant differences between groups.

**Continuous Variable**	**N**	**Median**	**Statistic**	***p*****-Value**	**-**	**-**
Length of stay	SH: 110 GH: 110	24.50 26.00	W = 12416.00	0.58		
**Categorical Variables**	**N**	**Percent**	**Statistic**	***p*****-Value**	**Odds Ratio**	**95% CI**
Live release	SH: 110 GH: 110	96.36 92.73	χ^2^ = 1.41	0.24	2.08	0.61–7.12
Return	SH: 104 GH: 93	16.36 31.81	χ^2^ = 2.32	0.13	0.54	0.25–1.20

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
