# Peer review of "A Comparison of Cats (Felis silvestris catus) Housed in Groups and Single Cages at a Shelter: A Retrospective Matched Cohort Study"

_animals, 2018, doi:10.3390/ani8020029_

Round 1
Reviewer 1 Report
Dear Authors,
This is an interesting study, well written and organized, on an important topic, SH vs GH, that is in need of more research. I have several concerns about the manuscript I would like to see addressed prior to recommending publication.
1) Cohort studies are typically prospective, please state that this is a retrospective or historical cohort study. Were any changes in protocols implemented during this period? How did you control for these? Where cats matched for the time (date/season) they were in the shelter? How about cat density in the shelter? If there were more cats say in June than in Nov, in theory the staff could have more time to devote to each cat, and the cat could have a shorter stay simply b/c there are fewer cats for potential adopters to choose from? More info needed here.
Data analysis: Typically in cohort studies the obvious analysis is to look for associations using relative risk or odds ratios- why was this analysis not done in this study?
2) Was any type of sample size calculation performed a priori? This is a relatively small N so lack of significant results could be due to lack of power. Please add calculations to M &M.
3) While you discuss in the introduction the pros and cons of SH vs GH there is no mention of the natural history of the cat. Cats' natural social group is matrilineal, males typically disperse from the natal territory, and while behaviorally flexible, they are not gregarious or a truly social species. All of these factors make living in unstable social groups in a shelter potentially very problematic. Until we understand how this type of housing impacts their behavior and welfare all other studies about shelter outcomes are beside the point. In your intro you reference a study that found GH increases solitary play not social play so perhaps it's the larger, enriched room, and the perception of more control rather than GH per se that is changing their behavior. I think this introduction could be stronger- see Ramos et al, 2013; Bradshaw and Hall, 1999
4) The SH and GH were wildly different making comparisons of the cats' experiences difficult. SH was NOT enriched- providing a toy is not enrichment, bare minimum should include hiding and perching opportunities which the GH cats had. Provision of hide/perch opportunities decreases the cats' perception of threat therefore increasing the possibility of play behavior. So comparing these very different environments and the behavior or outcomes of the cats in them is problematic. Also, while minimum cage size has not been been definitively described, 0.2 m2 is extremely small with 0.6 m2 the agreed upon minimum norm and having compartments is much different than not so all this is confounding. While this didn't appear to make a difference in your results the different types of stressors could be canceling each other out; social stress in GH and small, barren cages in SH making neither option as implemented "welfare" friendly for the cat.
5) A better job of describing the shelter husbandry procedures and daily interactions is needed. Were they identical for the 2 groups? Were any changes implemented over the course of the of the period reviewed?
6) Line 274-276- Social group instability is a social stressor for all animals so I disagree with this statement- If you mean the cat being moved is not that stressed b/c they are not socially bonded then doesn't that weaken the argument for group housing cats in shelters? And the remainder of the group will be impacted.
7) Line 293-294 it takes most cats at least 24 hrs to acclimate see Stella et al 2014, 2017. Therefore "short" moves are likely potent acute stressors and should be avoided.
8) Line 297-299 This isn't how socialization works- if cats are socialized to people as kittens and have a more gregarious personality than they will be more likely to interact with people but not necessarily unfamiliar people. And cats in SH, if provided hide boxes, will also be able to retreat and move away from people
9) Line 303 Your SH housing was NOT similar to Gourkow's which provided a hide/perch for the cat. Your housing was a barren cage, from the cats' point of view.
10) Line 310 should be approximately
11) Line 322 perhaps a tolerant relationship is better terminology- Unrelated adult cats may coexist but not necessarily be "friendly" toward each other
12) Line 334 should be contributions
13) Line 354 please change to retrospective cohort
Reviewer 2 Report
Overall, this is a comprehensive piece of work with a very clear rationale and methodology. Limitations of the findings are clear and the discussion matches the data. It is very clearly written to aid readers in understanding the work. I have only the following minor suggestions to make:
88-95 I suggest including Foster & Ijichi (2017) here to balance your discussion because we found that single housed cats may be more stressed and aid your rationale that more research is needed to address contradicting evidence. This is only a suggestion however.
131-132 Please state the sample for both groups- although this is mentioned later is isn't really clearly stated for the reader
Within the results, please include the sample size for each test within the brackets as missing data points will cause this to vary with each test and is rarely the full sample.
For each graph, include the statistical results in brackets as they are within the text. This gives your graphs greater impact and means they can stand alone
The opening paragraph of the discussion dives into results and conclusions too quickly. Start with a quick reminder of the rationale/importance of the work, then the knowledge gap you sought to address and overview of how you did that. Finish with just a teaser of your results and avoid the early conclusions you've included here (263-267). Save those for later in the discussion.
290-291 you have "case" twice
310 Approximately (not approximate)
Author Response
Please see the attached file with our response.

Reviewer 3 Report
Summary
The aim of this study was to use retrospective data to create two cohorts of cats, matched by signalment, but who experienced either single or group housing in a shelter and compare outcomes. These included moves in the shelter, length of stay, and outcomes. Overall they found that the two types of housing were fairly equivalent. More group housed cats were returned to the shelter and more single housed cats were moved off the shelter premises. While this paper has information that I believe is valuable to the body of knowledge on housing and movement of cats in a shelter setting, I believe it has many limitations.
Broad comments:
Introduction
- Good review of relevant previous studies and supporting material
Materials and methods:
- See below for specific comments about the housing types.
- Did you exclude or include cats that were readmitted to the shelter during your time period? Are these only cats who are having their initial intake? This might greatly affect their LOS and outcome as you note later in the paper cats that return to the shelter have a decreased positive outcome. You should exclude cats that were returned to the shelter from the initial cohorts and state this.
- You discuss spay/neuter. This can greatly affect LOS for cats if they are waiting for surgery they may not be available for adoption or able to go home. If in this shelter spay/neuter status does not affect LOS you should state this and explain why.
- You discuss moves to foster as having a 0/1 outcome in your analysis. Was time spent in foster included in your LOS calculation? Again a cat not on site or visible for adoption will have an increased LOS compared to those available. Is it a fair comparison?
- In outcome measures, the returns is unclear. I think you mean to say that any cats that left through a positive outcome were assessed to see if they were returned to the shelter in a 2 year time frame but the way this is described does not make sense.
- It does not affect your analysis but consider calling your analysis Wilcoxon rank-sum as opposed to Mann-Whitney U tests as this may be more recognizable in this field. Would it not be more appropriate to use a Wilcoxon signed-rank test as you matched your cohorts on several biological factors?
- Is there a reason you did not look at correlation between your different variables? Perform regression modeling? Consider confounding?
Results:
- In several places line 237 and 253, you discuss the difference in occurrence as “___ times more likely” but you did not calculate odds or risk ratios, or if you did you do not describe this in your methods. Is it not more appropriate to just discuss this difference in proportion as a frequency and not a likelihood?
- It would be nice to see a table of the outcomes and the statistics as well as discussion in written format.
- Along with median it would be nice to see the interquartile ranges. Consider doing some in depth description of descriptive data before discussing your statistical analysis
Discussion
- Why was no analysis done for whether GH or SH cats had more moves to isolation? Moves to isolation could greatly affect LOS and I think a variable that should have been investigated further.
- What was the shelters live release rate?
- Is 24 days LOS good or bad? Is that industry standard? I think that a discussion of this is necessary.
- There is no discussion of the limitations of the statistics?
Specific comments:
Introduction:
- You reference the Newbury shelter guidelines later in the paper but these include many of the concepts you discuss first thing including housing features for welfare
- Line 54 – “cats with more enrichment due to the having more space”
- Line 61 – need reference to [11] at the end of the sentence
- Line 91 – need references to the “several studies” added here
- Line 97 – “could be due to many reasons including”
- Line 99 – “pertaining to the husbandry”
- Line 106 – “Brown and Morgan”
- Lines 114-115 – could you find a way to rephrase? Perhaps “Behavioral measures of stress were not focused on in this study for the following reasons…”
- Lines 119-121 – rewrite. This is poorly phrased
Materials and methods:
- When describing GH, is it all in all out or are cats added to existing groups as then enter the shelter?
- Lines 153-159 – I am not entirely sure if this description is even necessary. You discuss Isolation here but never discuss moves to isolation in the rest of the paper also this is single housing so if group housed cats are moved to isolation are they still considered GH cats? This does not seem accurate. Again, does time spent in offsite locations add to LOS, does the type of housing they are moved to their change their categorization? You don’t discuss on site vs off site adoptions and this seems to be an important factor when you think of the different types of housing.
- Line 185 – “Cats who were present longer at the shelter”
- While reporting outcomes at 1/0 is something I understand, this may not be intuitive to all readers
Results:
- See above
Discussion
- 290-291 poorly stated
- Lines 345 – make the reference a separate ()
Round 2
Reviewer 1 Report
Thank you for responding to all my comments. This will be a nice addition to the literature
Reviewer 3 Report
Thank you, you did a very thorough job of answering all of my questions. I find your results to be very interesting and definitely contributing to the body of literature we have on cats, cat housing, and outcomes in shelters. Very excellent work.
A few very minor edits in the new content.
Line 32 - "Length of stay, and, live release rates, and returns after adoption did not
Line 321 - "were more likely"
Line 346 - "such as differences in shelter management"
Line 390 - "demonstrates one of the tradeoffs shelters face when"
Line 394 - " if the cat is stressed out by the"